# New Routine NLTE15$\mu$mCool-E v1.0 for Calculating the non-LTE CO$_2$ 15 $\mu$m Cooling in GCMs of Earth's atmosphere

Alexander Kutepov[1] and Artem Feofilov[2]

[1]The Catholic University of America, Washington, D.C.
[2]LMD/IPSL, Sorbonne Université, UPMC Univ Paris 06, CNRS, École polytechnique, Palaiseau, 91128, France

**Correspondence:** Alexander Kutepov (kutepov@cua.edu)

**Abstract.** We present a new routine for calculating the non-local thermodynamic equilibrium (non-LTE) 15 $\mu$m CO$_2$ cooling/heating of mesosphere and lower thermosphere in General Circulation Models. It uses the optimized models of the non-LTE in CO$_2$ for day and night conditions and delivered cooling/heating with an error not exceeding 1 K/day even for strong temperature disturbances. The routine uses the Accelerated Lambda Iteration and Opacity Distribution Function techniques for the exact solution of the non-LTE problem and is about 1000 faster than the standard matrix/line-by-line solution. It has an interface for feed-backs from the model and is ready for implementation. It may use any quenching rate coefficient of the CO$_2$($\nu_2$)+O($^3$P) reaction, handles large variations of O($^3$P) and allows the user to vary the number of vibrational levels and bands to find a balance between the calculation speed and accuracy. The suggested routine can handle the broad variation of CO$_2$ both below and above the current volume mixing ratio, up to 4000 ppmv. This allows using this routine for modeling the Earth's ancient atmospheres and the climate changes caused by increasing CO$_2$.

## 1 Introduction

The infra-red radiative cooling of atmosphere is an important component of its energy budget, which requires special techniques for its reliable estimation. For the local thermodynamic equilibrium (LTE), when the molecular emissions of the atmospheric unit volume are described by the Planck function for local temperature, this estimation confines on the solution of the radiative transfer equation in broad spectral regions occupied by molecular bands. In the middle and upper atmosphere, the breakdown of LTE (non-LTE) requires finding the sources of the non-equilibrium molecular emissions, which are obtained by the solution of the non-LTE problem. This makes the non-LTE cooling calculation significantly more time consuming. Today, the common opinion is that it is impossible to use exact methods for calculating the radiative cooling in GCMs, see, for instance, (López-Puertas et al., 2023) who promote this point of view. Various parameterizations were developed for fast calculating this cooling in GCMs, see (Fomichev et al., 1998; Fomichev, 2009), and (Feofilov and Kutepov, 2012) for review of works on parameterizing the non-LTE cooling of the Earth's middle atmosphere in the 15 $\mu$m CO$_2$, 9.6 $\mu$m O$_3$ and rotational H$_2$O bands. These algorithms, however, lack the accuracy needed for current GCMs.

Infrared emission in the 15 $\mu$m CO$_2$ band is the main cooling mechanism of middle and upper atmospheres of Earth, Venus and Mars (e.g. Goody and Yung, 1995; Sharma and Wintersteiner, 1990; Pollock et al., 1993; Bougher et al., 1994; Lopez-

Puertas and Taylor, 2001; Feofilov and Kutepov, 2012) On Earth, the magnitude of MLT cooling affects both the mesospheric temperature and height: the stronger the cooling, the colder and higher is the mesopause (Bougher et al., 1994).

The goal of this work is to present a new routine for calculating the non-LTE 15 $\mu$m $CO_2$ cooling of the Earth's mesosphere and lower thermosphere (MLT), which utilizes exact radiative transfer and non-LTE problem solution techniques and is fast enough to be applied in GCMs. The new routine can handle the broad variation of $CO_2$ both below and above the current

volume mixing ratio, up to 4000 ppmv. This allows using this routine for modeling the Earth's ancient atmospheres and the climate changes caused by increasing $CO_2$.

The routine we present is the optimized version of the research ALI-ARMS (for Accelerated Lambda Iteration for Atmospheric Radiation and Molecular Spectra) model and code (Kutepov et al., 1998; Feofilov and Kutepov, 2012). In this code the atmospheric cooling is the by-product of the non-LTE problem solution. For nearly 20 year the earlier version of this routine

is being successfully applied (Hartogh et al., 2005; Medvedev et al., 2015) in the GCM of Martian atmosphere.

We demonstrate in this paper the performance of the routine for calculation of the 15 $\mu$m $CO_2$ cooling of the Earth's MLT. However, generally this is a universal code, which potentially may be applied to modeling the non-LTE cooling in any molecular band or bands of several molecules in any planetary atmosphere. We successfully tested it calculating the non-LTE cooling of the Earth's MLT in the 6.3 $\mu$m $H_2O$ (Feofilov et al., 2009) and 9.6 and 4.7 $\mu$m $O_3$ (Manuilova et al., 1998) as

well as the cooling of Titan atmosphere in the 6.7 and 3.3 $\mu$m $CH_4$ bands (Kutepov et al., 2013; Feofilov et al., 2016). This potentially allowed applying this algorithm for calculating the radiative cooling in GCMs of various atmospheres, among them the atmospheres of water reach exoplanets (e.g. Valencia et al., 2007; Acuña et al., 2021) and gas giants.

In the next section, we outline techniques applied currently in GCMs for calculating the non-LTE $CO_2$ 15 $\mu$m cooling. In section 3 we briefly discuss method and techniques applied for calculating this cooling in our new routine. In section 4, we

present the model of non-LTE in $CO_2$ used in the routine and the routine computational performance. section 5 discusses the accuracy of KF23 compared to the reference calculations. The Conclusion summarizes the results of our study. Appendix A contains technical details of the code and recommendations for its implementing and usage in GCMs.

## 2 The non-LTE radiative cooling of atmosphere and its calculations in General Circulation Models

The energy loss of the atmospheric unit volume due to the infra-red radiation is calculated as the radiative flux divergence taken

with the opposite sign:

$$h = -\frac{1}{4\pi} \int d\omega \int d\nu \frac{dI_{\nu\omega}}{ds} \qquad (1)$$

where $I_{\mu\nu}$ is the intensity of radiation at the frequency $\nu$ along the ray $\omega$,

With the local thermodynamic equilibrium (LTE), the discretization of integral radiative transfer equation (RTE) (e.g. Goody, 1964; Goody and Yung, 1995) leads to a simple linear algebra operation for calculating the LTE 15 $\mu$m band cooling:

$h = W \times B$ $\qquad (2)$

where $h$ is the vector of cooling in $N_D$ grid points, $B$ is the vector of Planck function for a local $T$, and $W$ is $N_D \times N_D$ matrix, which accounts for radiative transfer in a number of 15 $\mu$m $CO_2$ bands contributing to the total cooling.

Extension of General Circulation Models (GCMs) on the mesosphere and thermosphere required accounting for the non-LTE for calculating the $CO_2$ 15 $\mu$m cooling. The standard way of solving the non-LTE problem (e.g. Curtis and Goody, 1956; Goody, 1964; Goody and Yung, 1995) requires inverting $(N_L \times N_D) \times (N_L \times N_D)$ matrix, where $N_L$ is the number of $CO_2$ vibrational levels included in the model. The equation (2) in this case remains unchanged, however, the matrix $W$ is rebuilt to account for differences between the Planck function and the non-LTE source functions in each band resulting from the non-LTE problem solution. This makes the calculation of the non-LTE cooling dramatically costlier (see for more details section 3).

Fomichev et al. (1998) and Fomichev (2009) (see references therein) discussed in detail various routines, which were suggested in previous studies for calculating the 15 $\mu$m cooling in GCMs with accounting for the non-LTE. The direct matrix solution of simplified non-LTE problem in $CO_2$ (often with an approximate radiative transfer treatment) or using pre-calculated $W$ matrices for a limited number of atmospheric situations for further interpolation of its elements helped to reduce the computational time at the expense of calculation accuracy. The low computational efficiency kept the accurate accounting for the non-LTE in GCM difficult those days.

A new way of calculating the non-LTE 15 $\mu$m $CO_2$ cooling in GCMs was suggested by Kutepov (1978). Relying on results of Kutepov and Shved (1978) who showed that the fundamental 15 $\mu$m $CO_2$ band 01101 - 00001 (see below Figure 1) of main $CO_2$ isotope dominates the cooling above about 85 km, he derived the recursive expression for $h$ coming from the analytical solution of the first-order differential equation for the non-LTE cooling $h$ in the fundamental band. This expression directly accounted for the $CO_2(\nu_2)$+O($^3$P) quenching rate coefficient $k$ and the O($^3$P) density. It was derived using the "second-order escape probability" approach (Frisch and Frisch, 1975; Frisch, 2022) for the approximate solution of the Wiener-Hopf type integral radiative transfer equation in the semi-infinite atmosphere. The algorithm of Kutepov (1978), see its refined version by Kutepov and Fomichev (1993), calculates $h$ upward in the non-LTE layers using as a lower boundary condition the LTE $h$ or the non-LTE $h$ obtained using other technique. Fomichev et al. (1993) linked this algorithm to the matrix routine for calculating $h$ in the LTE layers developed by Akmaev and Shved (1982). Later Fomichev et al. (1998) modified the routine of Fomichev et al. (1993) by adding the interpolation of the $W$ matrices for the $CO_2$ within 150–720 ppmv using tables of pre-calculated elements and described in detail the structure of revised matrix $W$. These authors also extended the routine altitude range to layers above 110 km. Cooling in this region is calculated from the simple balance equation for the first excited vibrational level of main $CO_2$ isotope with accounting for absorption of the radiative flux from below, cooling-to-space and collisional quenching. For smooth temperature profiles, Fomichev et al. (1998) reported the maximal cooling calculation errors of less than 2 and up to 5 K/day, for 360 and 720 ppmv $CO_2$, respectively. In this paper we will call this routine F98.

Basic features of F98, namely (a) broad altitude range covered, (b) straightforward accounting for the non-LTE, (c) high computational efficiency attracted many users. For more than two decades F98 routine has been the most widely used algorithm for calculating the 15 $\mu$m $CO_2$ cooling in GCMs of mesosphere and thermosphere, see, for instance, (Eckermann, 2023) for its latest application. However, as we show below, the F98 errors are large for non-smooth temperature profiles reaching 20-25 K/day in the mesopause region. On the other hand, even very minor variation of the $CO_2$ cooling may have significant impact

on the GCM results in the Mesosphere and Lower Thermosphere (MLT). Kutepov et al. (2013b) showed (Figure 18.2) that variation of the $CO_2$ cooling of $\sim$ 1-3 K/day in the Leibniz-Institute middle atmosphere model (LIMA) (Berger, 2008) in the mesopause region caused significant warming up to 5-6 K (about 105 km) and cooling up to -10 K (below 105 km) at latitudes between 90°S and 40°N for July 2005. This and other test lead to the conclusion (Berger, private communications), that the
accuracy of cooling/heating rate calculations in GCMs "should not exceed 1 K/day for any temperature distribution".

Fortunately, this accuracy requirement overlapped in time with the dramatic progress in the non-LTE radiative transfer calculations. This allowed developing a new routine NLTE15$\mu$mCool-E (hereafter KF23 for brevity within this manuscript) for calculating the non-LTE 15 $\mu$m $CO_2$ cooling in GCMs of the Earth's atmosphere, which exploits new exact algorithms for solving the non-LTE problem and, therefore, fits enhanced accuracy requirements. At the same time, it is fast enough to be used in GCMs.

KF23 is the optimized version of our basic ALI-ARMS model and code (Kutepov et al., 1998; Feofilov and Kutepov, 2012), which utilizes two advanced techniques: (1) the ALI technique for the solution of the non-LTE problem, and (2) the opacity distribution function (ODF) technique for optimizing the radiative transfer calculations. In this section, we outline these techniques and the current status and latest applications of ALI-ARMS code.

## 3   Method and techniques applied in a new routine for calculating the non-LTE cooling

### 3.1   Solution of the non-LTE problem

The non-LTE problem has two primary constituents: (1) the statistical equilibrium equations (SEE), which express the equality of the total population and de-population rates for each molecular level; (2) the radiative transfer equation (RTE) which relates the radiation field to the populations of levels, at *all altitudes* in the atmosphere (Hubeny and Mihalas, 2015). Hence, the system of equations for the level populations is *nonlocal* (and nonlinear). The most obvious way of dealing with this situation is to iterate between the SEE and RTE. This process, traditionally called "lambda iteration" (LI), has been investigated in the astronomical context since 1920's (Unsoeld, 1968). It inverts $N_D$ matrices $N_L \times N_L$ at each iteration step. This simple approach has been from time applied in the Earth's and planetary atmosphere radiative transfer, see, for instance, (e.g. Appleby, 1990; Wintersteiner et al., 1992). If the optical depths are large (as it is the case for the $CO_2$ 15 $\mu$m band) the algorithm converges slowly. Kutepov et al. (1998) studied several LI schemes and showed that for the $CO_2$ non-LTE problem, the number of iterations $I_{LI}$ for these algorithms may reach $\sim$ 200 for even moderate convergence criterion 1e-3. This results from the photons trapped in the cores of the most optically thick lines and of the strong SEE non-linearity related to quasi-resonant exchange of vibrational energy by the molecular collisions.

An alternative way of dealing with the non-LTE is a joint treatment of SEE and RTE, when RTE is discretized with respect to the optical depth or altitude grid to get a matrix representation of radiative terms in SEE. This approach, known in the atmospheric science as Curtis-matrix (CM) technique (e.g. Goody, 1964; Goody and Yung, 1995; Lopez-Puertas and Taylor, 2001)), leads to the matrix of dimension $(N_L \times N_D) \times (N_L \times N_D)$. In stellar atmosphere studies the generalized version of this technique is known as the Rybicki method (Mihalas, 1978). The time required for the solution of the non-LTE problem

using the CM technique is controlled by the number of operations for matrix inversion. The advantage of classic matrix method lies in the simultaneous determination of all populations at all altitudes instead of the sequential evaluation of populations step by step at each altitude using the radiative field from the previous iteration. Therefore, "matrix iteration" converges usually better than lambda iterations. However, the convergence of both algorithms depends strongly on how the local non-linearity is treated, see next section. To construct an adequate model, one must account for a large number of exited levels of various molecular species plus use a detailed model of atmospheric stratification. As a result both $N_L$ and $N_D$ can become very large. The dimensions of primary matrices are reduced by introducing various assumptions (for instance, the LTE assumption for rotational sub-levels as well as LTE in the groups of vibrational levels closely spaced in energy, etc, see also the discussion of GRANADA code in section 3.4). Nevertheless, usually the time to solve the non-LTE problem using CM method exceeds significantly the time when LI algorithm is applied to the same problem (see section 5 for more details).

In the 1990s, stellar astrophysicists have developed a family of powerful techniques, which utilize lambda iteration with an approximate (or accelerated) lambda operator (see, Rybicki and Hummer (1991, 1992) and references therein). In these so-called ALI techniques (for Accelerated Lambda Iteration) the integral lambda operator, which links the radiation intensity at a given point with source function at all points, is approximated by a local (or nearly local) operator. With a local operator, the largest matrices again, as in the LI case, have dimension $N_L \times N_L$. However, in this case, the convergence is rapid since most of the transfer in cores of the lines (described by the local part of lambda operators) cancels analytically and only the difference between exact and approximate radiative terms in the SEE is treated iterative. Kutepov et al. (1998) shoved that for the $CO_2$ non-LTE problem $I_{ALI} << I_{LI}$ (see section 5 for more details).

## 3.2 Treating the strong local non-linearity caused by intensive VV-exchange

Strong local non-linearity of the non-LTE radiative transfer problem in molecular bands caused by intensive near-resonant exchange of vibrational energy between molecules was studied by Kutepov et al. (1998). They showed that this non-linearity causes a dramatic deceleration of the convergence. Various schemes of additional "internal" iterations (without recalculating radiative excitation rates) aimed at adjusting populations of levels coupled by strong inter- and intramolecular vibrational-vibrational (VV) exchange, did not bring any help. To accelerate the convergence, Kutepov et al. (1998) suggested "decoupling" which utilizes the Avrett (1966) approach of treating the "source function equality in the line multiples". The SEE terms, which describe the VV coupling, depend on the products $n_v n_{v'}$, where $n_v$ is the population of vibrational level $v$ of one molecular specie, whereas $n_{v'}$ is the population of level $v'$ of the same or another molecular specie. In the iteration process, one needs to present these terms as $n_v n_{v'}^\dagger$, where $\dagger$ denotes the population of level with the *lower* degree of excitation, which is taken from the previous iteration. Kutepov et al. (1998) showed that this stops "the propagation of errors" by iterations and guarantee the fastest convergence. This decoupling requires only slight modification of matrices to be inverted (without additional linearization of the non-LTE problem and, therefore, additional programming efforts) provides, however, the same acceleration of convergence as the application of the Newton–Raphson method for the solution of system of non-linear equations (Gusev and Kutepov, 2003).

### 3.3 The radiative transfer in the molecular bands

With line-by-line (LBL) calculations, a very large number of frequency points to be accounted for significantly decelerate calculations the 15 $\mu$m $CO_2$ radiative fluxes. In LTE the reduction of frequency points is usually achieved by utilizing the so-called CKD (for correlated $k$-distribution) method that is based on grouping the gaseous spectral transmittances in accordance with the absorption coefficient $k$. The accuracy of this approach is better than 1%, see, for instance, (Fu and Liou, 1992). However, the $k$-correlation is not applicable under the non-LTE conditions because the vibrational level populations involved in the $k$-distributions are unknown and depend themselves on the solution of radiative transfer equation.

To overcome this problem, stellar astrophysicists developed the ODF technique. In this approach, they treat the non-LTE radiative transfer in "super-lines" associated with multiplets of very large line numbers (e.g. Hubeny and Lanz, 1995; Hubeny and Mihalas, 2015). They re-sample the normalized absorption and emission cross sections of "super-lines", consisting of hundreds or thousand of lines to yield a monotonic function of frequency that can be represented by relatively small numbers of frequency points. Though the idea is like the k-correlation, these normalized absorption and emission profiles do not depend on the total populations of upper and lower "super-levels", but only on the relative population of closely spaced in energy sublevels within each "super-level", which are supposed to be in LTE.

Feofilov and Kutepov (2012) described the adaptation of the ODF technique to the solution of the $CO_2$ non-LTE problem. They treated each $CO_2$ band branch as "superline". Therefore, each $CO_2$ band was presented by only 3 "lines" for perpendicular- and 2 "lines" for parallel bands. This way of treating the radiative transfer in the molecular band is about 50-100 times faster than the classic LBL approach. Whereas in LBL approach the radiation transfer equation is solved for each of $N_F$ frequency grid points within each rotational-vibrational line, in the ODF techniques the same number of frequency grid points as applied only to each "superline". Thus, the acceleration factor is approximately equal to the number of rotational-vibrational lines in the branch. As Feofilov and Kutepov (2012) show, the ODF approach introduces very small errors in the 15 $\mu$m cooling. For current $CO_2$ density ( 400 ppm in the lower atmosphere), these errors do not exceed 0.3 K/Day in a broad range of temperature variations, see Figure 18 of (Feofilov and Kutepov, 2012). They increase roughly linearly with the $CO_2$ increase.

### 3.4 From matrix and LI to ALI technique

Since the 1960s the Curits Matrix algorithms, with the rare exceptions for LI mentioned above in section 3.1, have been dominating the solution of the non-LTE problems in the Earth's and planetary atmosphere including the studies of the the 15 $\mu$m $CO_2$ cooling (see, for instance, the work by Zhu (1990), who developed very advanced for that time Curtis Matrix parameterizations of $CO_2$ cooling of MLT). Numerous non-LTE studies of the research group from the Institute of Astrophysics in Granada applied the GRANADA (for Generic RAdiative traNsfer AnD non-LTE population algorithm) code described by Funke et al. (2012). The core of it is a standard CM algorithm, which the authors in this and their earlier publications prefer to called MCM (for Modified Curtis Matrix). The paper discusses various ways of splitting large matrices into blocks, solving the non-LTE problem for selected sub-sets of levels and iterating to get the solution for all vibrational levels. In stellar astrophysics

(Mihalas, 1978) this approach is known as the "generalized equivalent two-level approach for multi-level problems". For many years it has no use because of convergence problems. The GRANADA code also includes the LI, but not the ALI technique, although the transformation of LI into the ALI requires a minimum of programming efforts, but speeds up the convergence for optically thick problems at least 10 times (e.g. Rybicki and Hummer, 1991; Kutepov et al., 1998). Additionally, Funke et al. (2012) neither compared the computational performance of MCM and LI algorithms nor described the handling of a strong

local non-linearity caused by VV-coupling.

### 3.5   The ALI-ARMS code

Kutepov et al. (1991, 1997) successfully applied the ALI technique to study the non-LTE emissions of molecular gases in planetary atmospheres (the 4.3 $\mu$m $CO_2$ band in the Martian atmosphere, and the 4.7 $\mu$m CO band in the Earth's atmosphere, respectively). Kutepov et al. (1998) and Gusev and Kutepov (2003) described in detail the adaptation of the ALI code developed

by Rybicki and Hummer (1991) for stellar atmospheres to the solution of the non-LTE problem for molecular bands of planetary atmospheres. They studied the performance of new the ALI-ARMS code, and demonstrated computational superiority of ALI-ARMS compared to various LI and CM/MCM techniques. The ALI-ARMS and its applications were described by Feofilov and Kutepov (2012). Later it was applied to study the rotational non-LTE in the $CO_2$ 4.3 $\mu$m band in Martian Atmosphere, observed by the Planetary Fourier Spectrometer (PFS) in the Mars Express mission ((Kutepov et al., 2017) and references therein), to

self-consistent two-channel $CO_2$/temperature retrievals from the limb radiance measured by the SABER (for Sounding of the Atmosphere using the Broadband Emission Radiometer) instrument on board of the Thermosphere Ionosphere Mesosphere Energetics and Dynamics (TIMED) Mission(Rezac et al., 2015), and for explaining the SABER nighttime $CO_2$ 4.3 $\mu$m limb emission enhancement caused by recently discovered new channel of energy transfer from OH($\nu$) to $CO_2$. Earlier, the ALI-ARMS code was used (Kutepov et al., 2006) to pinpoint an important missing process of strong V-V coupling between the

isotopes in the $CO_2$ non-LTE model of the SABER operational algorithm, and to modeling the $H_2O$ 6.3 $\mu$m emission and the $H_2O$ density retrievals in the MLT from the SABER 6.3 $\mu$m limb radiances (Feofilov et al., 2009). As we show below in sections 4 and 5, the ALI-ARMS code also provides an efficient way of calculating the 15 $\mu$m $CO_2$ cooling/heating in the GCMs.

## 4   New routine for the $CO_2$ cooling calculations

### 4.1   The $CO_2$ non-LTE day- and night-time models

To optimize the cooling calculations we used as a reference one our working line-by-line non-LTE model in $CO_2$, which comprises 60 vibration level of 5 $CO_2$ isotopic species and two-levels $N_2$, $O_2$ and O($^3$P). In Figure 1, we show the lower levels of this model (up to 5000 cm$^{-1}$). The set of collisional rate coefficients for the vibrational-translational (VT) and VV exchanged we apply is described by Shved et al. (1998) and is similar to rates used by Lopez-Puertas and Taylor (2001).

However, it relies on different scaling rules based on the first-order perturbation theory. Compared to our extended line-by-line

model (Feofilov and Kutepov, 2012), which includes total about 350 vibrational levels of 7 $CO_2$ isotopes and over 200000 rotational-vibrational lines, the $CO_2$ cooling of the 60-level model differs by less than 0.05 K/day and 0.5 K/day for the $CO_2$ mixing ratios of 400 ppmv and 4000 ppmv, respectively, both for daytime and nighttime conditions and for any temperature profile.

In the next steps, we gradually reduced the number of levels and bands in the model to optimize the calculations, keeping the cooling rate errors smaller than 1 K/day compared to the reference model.

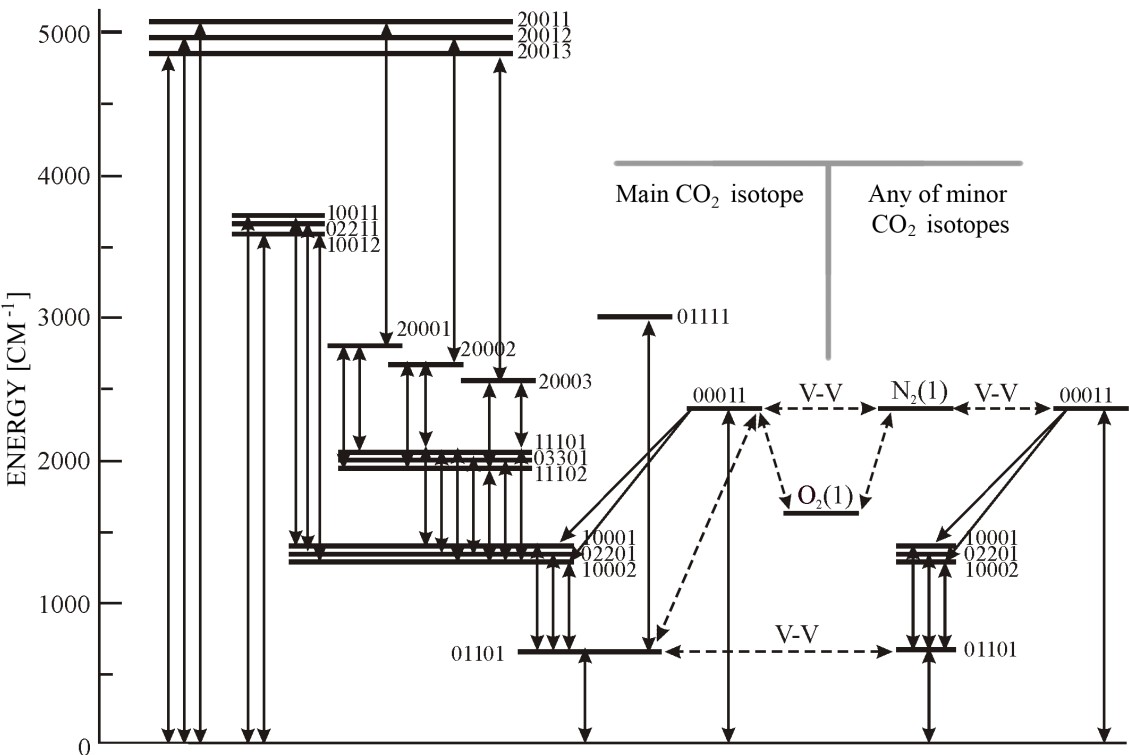

**Figure 1.** $CO_2$ vibrational levels diagram (Feofilov and Kutepov, 2012). We use the HITRAN (for *High*-resolution *Trans*mission Molecular Absorption database) notation for vibrational levels. Main optical transitions are shown by solid lines with arrows, dashed lines with arrows refer to the inter-molecular V–V energy exchange processes. V–T transitions are not shown for the sake of simplicity. The main $CO_2$ isotope levels are shown up to the 5,000 cm$^{-1}$ energy level. The minor isotopes levels are shown only up to 00011 level for simplicity.

**Table 1. CO₂ vibrational levels included in the night- and daytime model**

Isotope, HITRAN level notification, energy in cm-1

| | | | | | |
|---|---|---|---|---|---|
| 626 | 00001 | 0.0 | 636 | 00001 | 0.0 |
| 626 | 01101 | 667.379960 | 636 | 01101 | 648.478030 |
| 626 | 10002 | 1285.40834 | 636 | 02201 | 1297.26326 |
| 626 | 02201 | 1335.13161 | | | |
| 626 | 10001 | 1388.18432 | 628 | 00001 | 0.0 |
| 626 | 11102 | 1932.47013 | 628 | 01101 | 662.37335 |
| 626 | 03301 | 2003.24615 | | | |
| 626 | 11101 | 2076.85588 | 627 | 00001 | 0.0 |
| D 626 | 00011 | 2349.14291 | 627 | 01101 | 664.72941 |
| D 626 | 01111 | 3004.01227 | | | |
| D 626 | 10012 | 3612.84080 | 44 | 0 | 0.0 |
| D 626 | 02211 | 3659.27229 | D 44 | 1 | 2329.9116 |
| D 626 | 10011 | 3714.78193 | | | |
| D 626 | 20013 | 4853.62341 | 66 | 0 | 0.0 |
| D 626 | 20012 | 4977.83500 | D 66 | 1 | 1556.3519 |
| D 626 | 20011 | 5099.66050 | | | |
| | | | 6 | 0 | 0.0 |

D marks levels added in daytime model

In the Table 1, we show vibrational levels included in the optimized day- and nighttime models. The isotopes in the table are marked using the lower digit of the atomic weight: 626 corresponds to $^{16}O^{12}C^{16}O$, 636 corresponds to $^{16}O^{13}C^{16}O$ and so on. We account for 28 and 18 vibrational levels in the day- and nighttime models, respectively, which include the levels of four most abundant $CO_2$ isotopes and $N_2$ and $O_2$ levels (two and one for each at daytime and nighttime, respectively). We also include $O(^3P)$, but not $O(^1D)$ because its effect on the total $CO_2$ cooling/heating is negligible, both for nighttime and daytime conditions. The model uses the $CO_2$ spectroscopic information for all transitions available of the HITRAN-2016 (Gordon et al., 2017) for the levels listed in Table 1. In Table 2, we provide the numbers of bands, band branches, and lines used for daytime and nighttime calculations.

## 4.2 Computational performance and comparison with other algorithms

In the astronomical context, the detailed analysis of the operation numbers and times needed for the solution of the non-LTE problem is a must because of the complexity of the problem (Hubeny and Mihalas, 2015). Even though the number of levels and lines involved in the non-LTE problem of planetary atmospheres is smaller, the speed is a crucial parameter for the GCMs, so we perform the same type of analysis in this section below.

We compared three algorithms of the non-LTE problem solution, namely LI, ALI, and matrix method (hereafter MM), see for details section 2. Although LI technique is much more computationally expensive compared to ALI algorithms (Kutepov et al., 1998; Gusev and Kutepov, 2003), one cannot say the same about MM approach applied to a problem with a reduced number of levels because of the low number of iterations it usually requires, and this required testing. We generated the matrices of the MM-algorithm from the matrix presentations of lambda operators, as discussed by Kutepov et al. (1998). We used the discontinuous finite element (DFE) algorithm as the most efficient way of solving the radiative transfer equation (Gusev and

Kutepov, 2003; Hubeny and Mihalas, 2015), and we compared the LBL and ODF techniques (see section 2.3). Below in this section, we discuss the performances of all algorithms only with the non-linearity caused by near-resonance VV energy exchanges resolved as outlined in the section 3.2. Without this, the number of iterations of all considered algorithms would be a few times higher.

For each of the three techniques, we checked the numbers of operations and times needed for each single iteration, then accounted for a number of iterations and compared total numbers of operations and times for the entire non-LTE problem solution.

We found that the time required for each iteration of the algorithms we studied is dominated by 3 components: 1) time for auxiliary operations $T_{Aux}$ (such as filling of large arrays like vectors or matrices to be inverted), 2) time for solving the radiative transfer equation ($T_{Rad}$) and forming the radiative rate terms in the SEE, and 3) time for matrix inversions ($T_{Inv}$). The cooling itself is the by-product of the non-LTE problem solution and is estimated nearly instantaneously as in (Kutepov et al., 1998):

$$h = \sum_b \left[ \left( n_{lo} B_{lo,up} \overline{J_{lo,up}} - A_{up,lo} n_{lo} \right) \times (E_{up} - (E_{lo}) \right]_b \tag{3}$$

where $B_{lo,up}$ and $A_{up,lo}$, the band Einstein coefficients, and $\overline{J_{lo,up}}$, mean intensity in the band, which enter radiative rate coefficients of SEE matrices, whereas $n_{lo}/E_{lo}$ and $n_{up}/E_{up}$ are the populations/energies of lower and upper vibrational levels in the band, respectively. The sum in (3) goes along all $CO_2$ transitions in the model.

## Table 2. Numbers of operation and computing times.

**NIGHT: $N_L = 18$; $N_B = 18$, $N_{Br} = 54$, $N_{RT} = 3078$; $N_F = 32$; $N_A = 2$: $N_D = 125$; $CO_2 x1$**

|  | $N_{Aux}/T_{Aux}$ | $N_{Rad}/T_{Rad}$ | $N_{Inv}/T_{Inv}$ | $N_I$ | $T_{tot}$ | $K$ |
|---|---|---|---|---|---|---|
| MM/LBL | 5e4 / 9e-2 | 2.5e7 / 0.25 | 1.14e10/43 | 2 | 86 | 860 |
| LI / ALI, LBL | 5e4 / 8.5e-3 | 2.5e7 / 0.25 | 7.3e5/7.7.0e-3 | 60/5 | 16/1.3 | 160/ 13 |
| ALI, ODF | 5e4 / 8.5e-3 | 4.5e5 / 4.5e-3 | 7.3e5/7.7.0e-3 | 5 | 0.1 | 1 |

**DAY: $N_L = 28$; $N_B = 46$, $N_{Br} = 119$, $N_{RT} = 6039$; $N_F = 32$; $N_A = 2$: $N_D = 125$; $CO_2 x1$**

|  | $N_{Aux}/T_{Aux}$ | $N_{Rad}/T_{Rad}$ | $N_{Inv}/T_{Inv}$ | $N_I$ | $T_{tot}$ | $K$ |
|---|---|---|---|---|---|---|
| MM/LBL | 1.1e5/2e-1 | 4.8e7/0.48 | 4.3e10/160 | 2 | 321 | 1284 |
| LI / ALI, LBL | 1.1e5/1.9e-2 | 4.8e7/0.48 | 2.7e6/2.2e-2 | 60/5 | 31/2.6 | 124/10 |
| ALI, ODF | 1.1e5/1.9e-2 | 9.5e5/9.5e-3 | 2.7e6/2.2e-2 | 5 | 0,25 | 1 |

$T_{Aux}$, $T_{Rad}$, etc. are in s.

In Table 2, we present the summary of our study. The table gives the main parameters of the non-LTE models for day and night conditions described in section 4.1, operation numbers and the times in seconds (measured with the help of timing routine withing the code), which are required for each calculation part. We performed this study at two different machines, with x86_64

Intel and Intel Xeon Gold processors operating at 2.2 and 2.5 GHz, respectively. We compiled the ALI-ARMS code to be used in 64-bit architecture with the help of a standard gnu compiler collection (gcc) compiler and we ran it on a single processor. We provide the results only for one 2.2 GHz Intel processor; the timing for the second processor is roughly 1.4 times shorter.

Compared to the reference code, we ran the routine using the convergence criterion 1.0e-2 instead of 1.0e-4. This allowed reducing the number of iterations by a factor of about 2 without sacrificing the accuracy.

Similar to the study by Hubeny and Mihalas (2015), we found that time $T$ required for any procedure like the radiative transfer equation solution or matrix inversion may be presented as:

$$T = CN \tag{4}$$

where $N$ is the number of operations. Thus, whereas $N$ is defined by mathematical nature of the problem, and the algorithm applied the coefficient $C$ may depend on many factors like the quality of programming, language used, operational system, interpreter, computer architecture and performance, etc.

We found that the number of operations for the solution of the radiating transfer equation $N_{Rad}$ in case of the non-overlapping lines may be approximated by the expression:

$$N_{Rad} \simeq N_D \times N_{RT} \times N_F \times N_A \tag{5}$$

and is the same for all algorithms compared. Here $N_{RT}$ is the total numbers of lines (or band branches in the ODF case), and $N_F$ and $N_A$ are the numbers of points in the frequency and angle integrals used, respectively. The coefficient $C$ in (4) which links radiative transfer operation numbers and corresponding times was found to be $\simeq 1.0e-8$ s.

We found that with the LI/ALI algorithms, the number of auxiliary operations $N_{Aux}$ is well approximated by the expression:

$$N_{Aux}^{LI/ALI} \simeq N_L^2 \times N_D \tag{6}$$

where $N_L$ is the number of $CO_2$ vibrational levels. This expression gives the number of terms to be filled in the block-diagonal matrix comprising $N_D$ blocks $N_L \times N_L$, where $N_L$ is the number of vibrational levels. In case of the LI/ALI techniques these are the $N_D$ matrices generated and inverted one after another at each iteration step. The coefficient $C$ in (4) which links auxiliary operations and corresponding times is $C \simeq 1.7e-7$ s.

In case of MM technique the matrix to be generated at each iteration is much larger, namely it has the size $(N_L \times N_D) \times (N_L \times N_D)$ and consists of $N_L$ fully filled diagonal blocks $N_D \times N_D$ which represent non-local radiative terms, whereas the same as for LI/ALI case collisional terms are now spread over non-diagonal parts of this large matrix. We found that when we present the number of operations to fill this matrix as:

$$N_{Aux}^{MM} \simeq N_D^2 \times N_L \tag{7}$$

then approximately the same coefficient $C \simeq 1.7e-7$ s links this number with the time needed for its filling.

The number of operation needed to matrix inversion $N_{Inv}$ is approximately $N^3$, where $N$ is the matrix dimension. Thus, we have the following expressions

$$N_{Inv}^{MM} \simeq (N_L \times N_D)^3 \qquad (8)$$

for the MM algorithm, and

$$N_{Inv}^{LI/ALI} \simeq (N_L \times N_L)^3 \times N_D \qquad (9)$$

for the LI/ALI techniques. In the latter case the number of operations is $N_D^2$ times lower, since only $N_D$ matrices $N_L \times N_L$ are inverted one after another at each iteration. This is the great advantage of these techniques compared to MM when the entire huge matrix needs to be inverted at once since it has non-zero elements outside the diagonal blocks.

We use in the ALI-ARMS standard routines for linear equation solution (e.g. Press et al., 2002) ludcmp (LU decomposition), lubksb (backsubstitution) and mprove (iterative improvement) for matrix inversions. We found that for these routines applied to the non-LTE problems studied here the coefficient between the number of operations and time for matrix inversion depends on the matrix dimension $N$ and may be approximated by expression:

$$C = 1.0e\text{-}7 \cdot (0.04 + 1.2/N). \qquad (10)$$

One may see it the upper part of Table 2 that for night conditions the application of MM technique causes the matrix inversion be the most time-consuming calculation part at each iteration, although the number of iterations $N_{Iter} = 2$ is low. Applying LI,LBL techniques, provides strong reduction of the matrix inversion time per one iteration gives, however, only moderate reduction of total time (by a factor $\sim 5$) compared to MM,LBL due to a large number of iterations (60). The number of iterations for the LI/ALI techniques slightly depends on the atmospheric pressure/temperature distribution. The numbers, which are given in the table are mean values for a few hundreds of run for different atmospheric conditions. Applying ALI instead of LI significantly reduces the number iteration (5 instead of 60) providing additional acceleration of calculations by the factor $\gtrsim 10$. We note, that for the LI/ALI,LBL the most time-consuming part for each iteration is now the radiative transfer solution, which is more than 15 times slower than two other parts of calculation together. The ODF technique allows reducing $T_{Rad}$ by a factor 50-60. This provides total additional acceleration by a factor $\gtrsim 10$. The last column in table gives the acceleration factor $K = T_{tot}/T_{tot,ALI,ODF}$, which shows how faster the ALI,ODF combination works compared to other techniques: it is about 900 and 160 time faster than MM,LBL and LI,LBL techniques, respectively.

The lower part of Table 2 shows the number of operations and times for various parts of calculations for the daytime non-LTE model described in section 4.1. Compared to nighttime, daytime calculations require about 2.5 times more time due to increased number of vibrational levels and bands accounted for. Nevertheless, main points discussed above in this section for the nighttime runs remain valid for daytime: (a) main decelerating factor for MM technique is the matrix inversion, notwithstanding the low number of iterations; (b) LI technique although reduces the matrix inversion time by a factor $\gtrsim 7000$, provides, however, only moderate decrease of total time (by a factor of $\sim 10$) because of the large number of iterations; (c) the ALI technique is more over 100 times faster than MM with the slowest part of calculations to be the LBL solution of RTE; (d)

the ODF provides acceleration of RT calculations by a factor of 50. Finally, the ALI,ODF technique appears to be over 1000 times faster than the MM,LBL approach.

We measured the time the F98 routine requires at x86_64 Intel 2.2 GHz processors and found it to be around 3e-4 s. It means that for nighttime KF23 is about 300 times slower than F98. At daytime, when the solar heating parameterization of Ogibalov and Fomichev (2003)) is accounted for, our version of F98 requires about 30% more time. Still, it remains about 600 times faster than the daytime KF23 routine. In the next section we discuss in detail the accuracy of both routines. This study will help the user to choose between the accuracy and calculation speed depending on the errors his model may tolerate.

## 5    Accuracy of cooling/heating rate calculations

To estimate the calculations errors, we compared the outputs of our new KF23 routine and the F98 routine for the non-LTE $CO_2$ cooling calculations with our non-LTE reference model, discussed in section 4.1. In these tests, we used the $CO_2$ volume mixing ratio (VMR) profiles with 400 ppmv in their "well mixed" part. We also tested the same profiles multiplied by factors of 2, 4, and 10. For the $CO_2(\nu_2)+O(^3P)$ quenching rate we used temperature-dependent coefficient $k = 3.0 \times 10^{-12}$

$s^{-1}cm^3 \times \sqrt{(T/300)}$ (see section 5.4 for more details). As we described before in section 4.1, the vibrations levels and bands accounted for in KF23 keep its accuracy of $\sim 1$ K/day for any temperature profile, including those strongly disturbed by various tidal and gravity waves. Here, we show the errors of both routines compared to the reference model only for "wavy" temperature profiles. For mean profiles with a smooth structure, the errors of KF23 were about 0.1-0.3 K/day (for 400 ppmv of $CO_2$). For the F98 routine, the errors for smooth profiles were around 1-3 K/day confirming the results of Fomichev et al.

345    (1998).

### 5.1    The nighttime cooling/heating rates

In Figure 2, we show five typical temperature profiles, which demonstrate superposition of different meso-scale waves. These profiles, as well as corresponding pressure, $O(^3P)$ and $CO_2$ distribution (shown in Figure 3), and other constituents from the Whole Atmosphere Community Climate Model Version 6 (WACCM6) (Gettelman et al., 2019) runs were kindly provided by

Dan Marsh (private communications). For the 15 $\mu$m cooling calculations this model uses the F98 parameterization. We show below the calculation results for these atmospheric model inputs because WACCM is widely used by the community model. Generally, any pressure/temperature profiles disturbed by strong waves, give similar results, as we observed in our tests. These may be both p/T distributions generated by modern GCMs, like in our case here, or those retrieved from ground base or space observations, or artificial wavy p/T distributions.

Figure 4 shows the $CO_2$ 15 $\mu$m cooling rates for the new KF23 routine, for the F98 parameterization, and for our reference model for temperatures in Figure 2 and the 400 ppmv $CO_2$ profiles. One may see in this figure that errors of new routine do not exceed 0.5 K/day. On the other hand the F98 routine errors reach up to 13 K/day. The altitude range, where the F98 routine demonstrates significant errors, is broad starting just above the altitude of 60 km. In Figure 5 we show same as Figure 4 cooling rates and their differences, however, for the twice higher $CO_2$ of 800 ppmv in the "well mixed" range. We note here that both

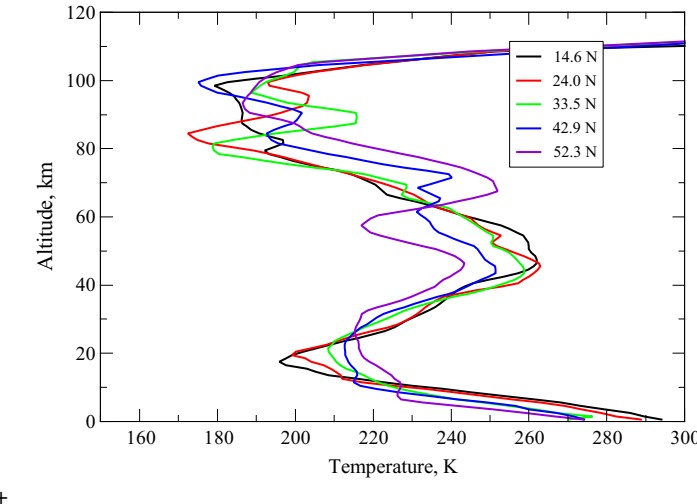

+

**Figure 2.** Temperature profiles used for testing the $CO_2$ 15 $\mu$m cooling calculations

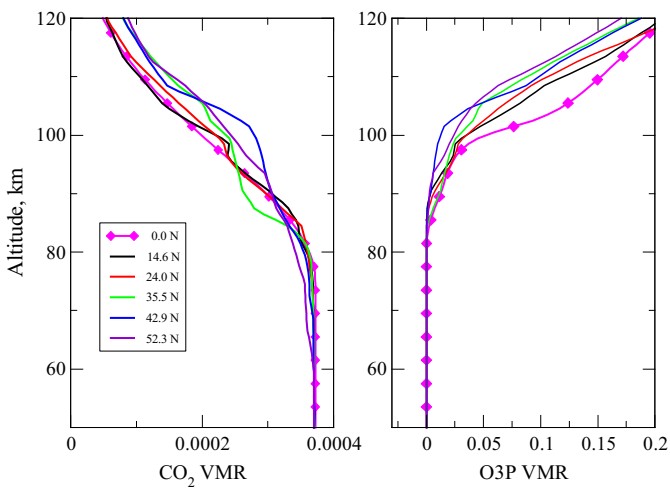

**Figure 3.** Volume mixing ratio profiles of $CO_2$ and $O(^3P)$ used for testing the $CO_2$ 15 $\mu$m cooling calculations. Solid magenta line with diamonds for 0.0 N are data taken from Yudin et al. (2022) which were used for the simulations shown in Figure 9; Solid lines for 14.6°N - 52.3°N correspond to the temperature profiles in Figure 2 and were used for simulations shown in Figures 4 - 6, see text for details.

.

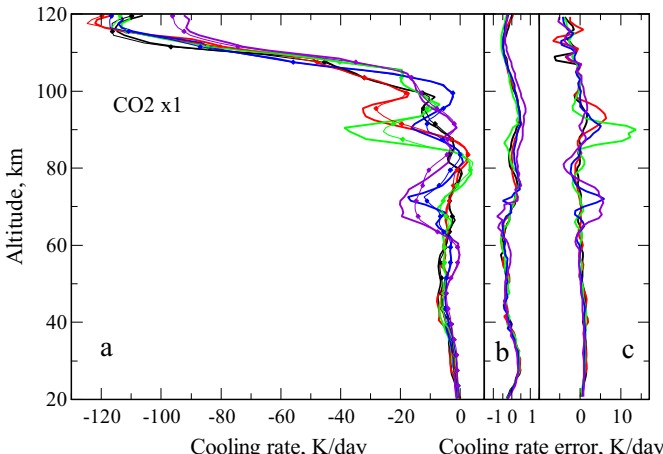

**Figure 4.** The nighttime cooling rates in the $CO_2$ 15 $\mu$m band and cooling rates errors for the $CO_2$ VMR of 400 ppmv for temperature distributions of Figure 2. Line colors correspond to the legend in Figure 2. (a) Cooling rates: thick solid lines - reference model; thin solid lines with diamonds - F98 routine; the KF23 results are not shown; (b) Cooling rate differences between the new routine KF23 and reference data; (c) Cooling rate differences between the F98 routine and reference data.

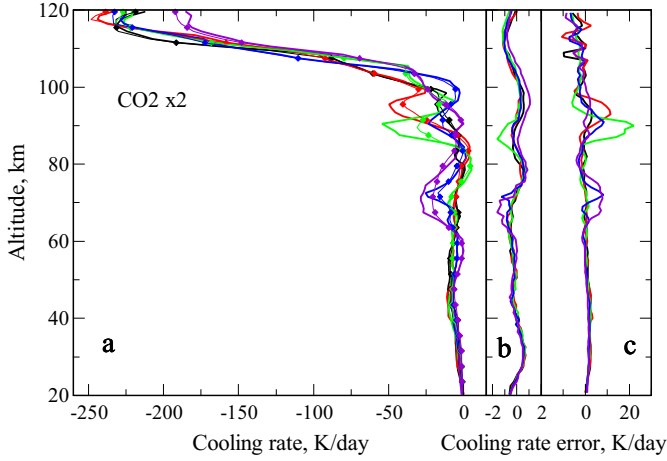

**Figure 5.** The same as in Figure 4, however, for 800 ppmv of $CO_2$.

maximal absolute values of cooling rates as well as the new- and the F98 routine errors are roughly twice higher compared to those of Figure 4. For the new routine they do not exceed 1 K/day, whereas for the F98 routine they reach up to 23 K/day.

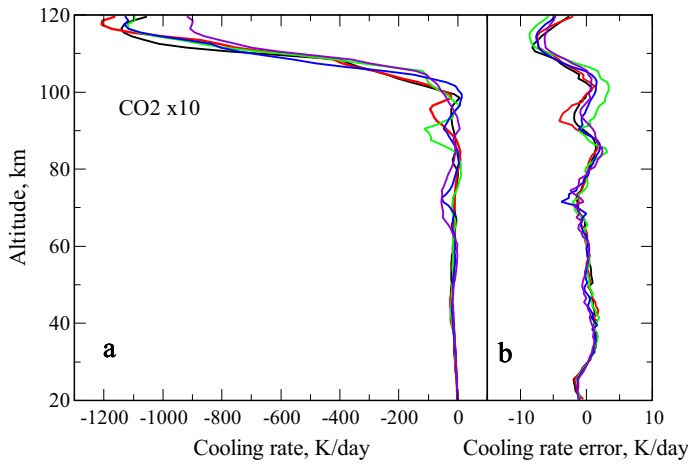

**Figure 6.** The same as in Figure 4, however, for 4000 ppmv of $CO_2$, no results for the F98 routine are shown.

Finally, in Figure 6 we show in panel (a) the cooling rates produced by the reference model and by KF23 routine for the $CO_2$ VMR of 4000 ppmv, which is 10 times higher then the reference one. The new routine errors are shown in panel (b). F98 routine was not tested at this inputs since it was not designed to work with the $CO_2$ VMRs higher than 720 ppmv. One may see

in this figure that absolute cooling rates (maximal values) are approximately 10 times higher than those for 400 ppmv of $CO_2$. Same is roughly true for the new routine errors, which now reach in upper parts of tested region the values up to 8 K/day.

In Figure 7 we show for night the contributions of major and minor $CO_2$ isotopes, included in the model (see section 4.1), for temperature at $33.5°N$ for 400 and 1600 ppmv of $CO_2$. One may see in this figure that for the 400 $CO_2$ ppmv this contribution does not exceed $\sim 2$ K/day and $\sim 1$ K/day for the 626 hot bands and for all minor isotope bands, respectively. This effect of hot

bands and minor species is increasing with the $CO_2$ density, see the right panel of Figure 7 particularly for altitudes affected by waves.

As we mentioned in the section 4.1 we use all $CO_2$ bands available in the HITRAN-2016 for the night set of levels in Table 1. This minimizes the errors compared to reference calculations to $\leq 1$ K/day for 400 ppmv of $CO_2$. The routine allows using less levels and bands to accelerate calculations, however, at the expense of error increase (see for more details Appendix A). For in-

stance, excluding (see Figure 1) weak first (10001,02201,10002)→01101 and second (11101,03301,11102)→(10001,02201,10002) hot bands of 626 and 636 isotopes makes the total number of bands twice lower. Our tests show that in this case the routine work only about 10% (see also Table 2) faster, however the maximal cooling rate error for 400 ppmv increases up to 3 K/day.

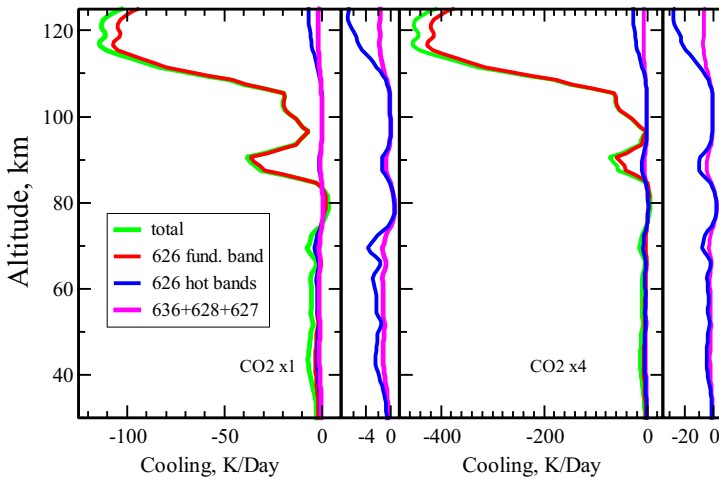

**Figure 7.** Contributions of various $CO_2$ bands to the total nighttime cooling at 33.5°N. The narrow sub-panels show re-scaled 626 isotope hot bands and minor isotope contributions.

## 5.2 The daytime cooling/heating rates

At daytime the near-infrared heating due to the absorption of solar radiation in the $CO_2$ bands around 2.0-4.3 $\mu$m represents a small, however, non-negligible reduction of the total $CO_2$ cooling (up to 1-2 K/day for the current $CO_2$). The complex mechanisms of the absorbed solar energy assimilation into a heat were investigated in detail in a number of studies summarized by Lopez-Puertas and Taylor (2001). Ogibalov and Fomichev (2003) studied this heating for smooth temperatures for various $CO_2$ and solar zenith angles (SZAs) and suggested the look-up-table, which allows a quick estimate of this heating in GCMs. Due to its reasonable accuracy ($\sim$ 0.5 K/day for current $CO_2$) this table has been used as a daytime supplement to the F98 nighttime cooling parameterization. Unfortunately, with increasing $CO_2$ density the errors of this table increase rapidly above $\sim$ 70 km: for 720 ppmv $CO_2$ it underestimates the heating around the mesopause by more than 50% (see Figure 5 of Ogibalov and Fomichev (2003) for daily averaged heating). In Figure 8 the heating of the atmosphere due to the daytime absorption of solar radiation in the $CO_2$ bands at 2.0-4.3 $\mu$m for SZA=45° at the latitude 33.5°N produced by our new routine is shown. To prevent increasing errors for daytime due to an inadequate treatment of solar radiation absorption and assimilation, the KF20223 utilises at daytime an extended non-LTE model which, compared to the nighttime, includes additionally 10 more $CO_2$ vibrations levels (see Table 1) and comprehensive system of radiative and collisional VT and VV energy exchanges as described by Shved et al. (1998); Ogibalov et al. (1998). Higher number of vibrational levels and more than twice higher number of bands leads to a 2.5-time longer time for the daytime cooling/heating calculation (see Table 2). However, the daytime errors were of the same order (less that 1 K/day for 400 ppmv) as those for the nighttime (Fig. 4-6) even for strongly perturbed

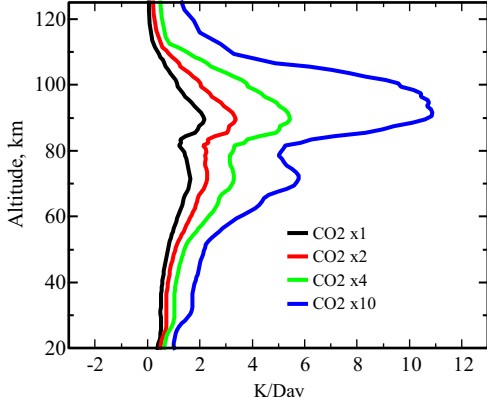

**Figure 8.** The near infra-red solar heating at 33.5° N for solar zenith angle 45° for various $CO_2$ densities.

temperatures and increased $CO_2$. We do not present these comparisons. As in the night case removing a half of bands (various hot and combinational bands) in the daytime model gives only about 10% speed gain, however, maximal errors of cooling may reach for 400 ppmv 4 K/day.

## 5.3 Cooling/heating rates for the diurnal tides at equator

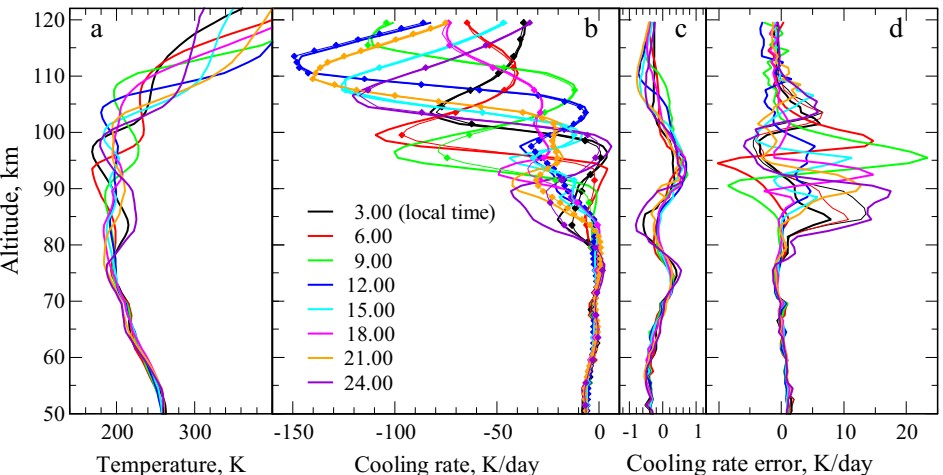

**Figure 9.** Comparisons of the $CO_2$ 15 $\mu$m cooling rates. Diurnal tides at equator. (a) Temperature profiles for various local times; (b) Cooling rates: thick solid lines - reference model, thin solid lines with diamonds - F98 routine; (c) Cooling rate differences between new routine, presented in this study, and the reference model; (d) Cooling rate differences between the F98 routine and the reference model

In Figure 9, we compare the cooling rates obtained with the KF23 routine and with F98 parameterization with those produced by the reference model. For these tests, we used the temperature/pressure distributions affected by the diurnal tides at the equator. $p$, $T$ and the atmospheric constituent densities as well as local zenith angles for local times at $1.0°$N and $12.0°$E (March 15, 2019) corresponds to the WACCM-X (Whole Atmosphere Community Climate Model extended in the thermosphere and ionosphere) equatorial simulations constrained by meteorological analyses of the NASA Goddard Earth Observing System version 5 (GEOS-5) below the stratopause, as discussed in Yudin et al. (2020, 2022). The temperature distributions at various local times are shown in panel (a). Panel (b) presents the $CO_2$ 15 $\mu$m cooling rates calculated using the reference model (thick solid lines) and the F98 routine (thin solid lines with diamonds). We do not show here the cooling rates obtained using the KF23 routine because the difference between them and the reference data does not exceed the thickness of a curve. These differences do not exceed 1 K/day (Fig. 9c). Figure 9d shows the differences between the cooling produced by the F98 routine and the reference calculations. In contrast to Figure 9c, these differences exceed 20 K/day for some temperature profiles.

One may see in Figure 9 that the accuracy of the F98 routine is improving above 100 km and below 80 km. Above 100 km, the F98 parameterization uses the recursive expression, the accuracy of which increases with height (Kutepov, 1978). Below 80 km, the F98 routine is based on LTE matrix algorithm for cooling calculations. This algorithm accounts for the radiative interaction of the neighboring levels and the escape of radiation to above, and, therefore, provides a good accuracy of cooling in this optically thick layer. In the layer between 80 and 100 km, the F98 routine merges the cooling values calculated by the two methods outlined above. This merging works reasonably well for smooth temperature distributions tested by Fomichev et al. (1998), but it fails with "wavy" temperatures. One also needs to keep in mind that this layer is the transition region between the LTE and non-LTE state of the $CO_2(\nu_2)$ vibrations, where the physics of formation of the non-equilibrium vibrational distribution must be considered in all details (e.g. Kutepov et al. (2006)). Compared to the KF23 routine, which rigorously models the non-LTE, the F98 routine fails in this situation, what Fomichev et al. (1993) warned about when they presented the first version of this parameterization.

## 5.4 The $CO_2(\nu_2)$+O($^3$P) quenching rate coefficient

The results above in section 5 were obtained for the temperature-dependent $CO_2(\nu_2)$+O($^3$P) quenching rate coefficient $k = 3.0 \times 10^{-12}$ s$^{-1}$cm$^3 \times \sqrt{(T/300)}$. The multiplier in this expression is the median value of the rate coefficient from the range of (1.5 - 6.0)$\times 10^{-12}$ s$^{-1}$cm$^3$ which spans from the low laboratory data up to high data obtained from the space observations of the $CO_2$ 15 $\mu$m emission, see (e.g. Feofilov et al., 2012) and references therein. This value is currently accepted for usage in the GCMs for calculation of the 15 $\mu$m cooling. The dependence of MLT cooling on $k$ was investigated in many previous works summarized by Lopez-Puertas and Taylor (2001), see also references therein). It is known that the maximum value of cooling, which is usually reached at the altitudes of 100-140 km, is roughly proportional to the $k$ value used in calculations. The KF23 routine works well for any $k$ from the range given above, and it also allows varying the temperature dependence of the rate coefficient (e.g. Castle et al. (2012)), see Appendix A. For our calculations, we used the O($^3$P) densities shown Figure 3.

## 5.5 Upper and lower boundaries

The accuracy tests of the KF23 routine were performed with the upper boundary of atmosphere at 130 km boundary. The routine may work with any upper boundary in the upper mesosphere and above. Putting the upper boundary below $\sim 110$ km may cause, however, increasing cooling errors due to not accounting for the upper atmospheric layers. The lower boundary can be placed at any altitude below $\sim 50$ km where all $CO_2$ 15 $\mu$m bands are in LTE. This will justify the LTE lower boundary condition for the radiative transfer equation solution. However, it is not recommended to use the routine results below $\sim 20$ km because of increasing errors by not accounting for the line overlapping in a current version of the ODF approach.

## 6 Conclusions

We present the new KF23 routine for calculating the non-LTE $CO_2$ 15 $\mu$m radiative cooling/heating in the middle and upper atmosphere. The routine provides high accuracy cooling rates above 20 km in a broad range of atmospheric input variations for any temperature distributions including those disturbed by strong micro- and meso-scale strictures and is the optimized version of the ALI-ARMS reference model and research code (Feofilov and Kutepov, 2012), which rigorously solves the non-LTE in $CO_2$, $N_2$, $O_2$ coupled by intensive vibrational-vibrational energy exchanges. The routine relies on advanced techniques of exact non-LTE problem solution (ALI-algorithm) and the molecular band radiative transfer treatment (ODF-technique). Using these methods, we achieved about $\sim 1000$ times acceleration compared to the standard matrix/line-by-line solution of the same non-LTE problem without sacrificing the accuracy. We show that the maximum error of calculations does not exceed 1 K/day for the current atmospheric $CO_2$ density and the median value of $CO_2(\nu_2)+O(^3P)$ quenching rate coefficient. This accuracy is ensured by a relatively large number of $CO_2$ levels and bands used in the KF23 routine. We also allow the user to choose between the accuracy and calculation speed by adding or removing certain bands and levels (see Appendix A).

The KF23 routine provides accurate cooling calculations in a vast range of the $CO_2(\nu_2)+O(^3P)$ quenching rate coefficient and $O(^3P)$ variations. It works also well for very broad variations of the $CO_2$ VMR, both below and above the current density, up to 4000 ppmv. Consequently, this allow to use this routine to model the Earth's ancient atmospheres and the climate changes caused by increasing $CO_2$.

Recently López-Puertas et al. (2023) presented an updated version of the F98 routine. Detailed analysis of this work is given by Kutepov (2023). The main improvement compared to F98 is an extended range of $CO_2$ abundances: whereas F98 routine covered the range of $CO_2$ concentrations with tropospheric values from 150 to 720 ppm the parameterization presented in this work goes up to 3000 ppm of tropospheric $CO_2$. Another minor improvement is the finer altitude grid of revised parameterization. The authors did not make revised routine available for users for its independent testing. They show numerous tests of the routine accuracy, however, only for undisturbed individual temperature distributions, for which its error does not exceed 0.5 K/day. They also tested the revised routine for the temperatures retrieved from the MIPAS (Michelson Interferometer for Passive Atmospheric Sounding) observations, which demonstrate large variability. These individual profiles are the good inputs for the revised parameterization to show how it works for strongly disturbed temperature profiles. However, the authors do not show these results. Instead, they present only zonal means of the differences. Obviously, this averaging washes out the

465 errors obtained for individual profiles, for which we observed (see section 5) the F98 parameterization errors up to 25 K/day. Meanwhile these large errors in our study are generally concentrated in the altitude region around 90 km, exactly where the Root Mean Square Errors (RMSEs) in Figure 25 of López-Puertas et al. (2023) are maximized reaching 8-9 K/day. In section 5 we explain why F98 works badly in this altitude region. These large RMSEs allow to conclude that presented by López-Puertas et al. (2023) revised routine has the same problems as the F98 parameterization and, therefore, works nothing better for the 470 disturbed temperature distributions then the F98 parameterization.

## Appendix A: The NLTE15$\mu$mCool-E v1.0 routine (technical details)

The routine source code is written in C. The routine is available at https://doi.org/10.5281/zenodo.8005028 and is ready for implementation into any General Circulation Models usually written in Fortran through a small "wrapper".

The routine has an interface, which allows efficiently receiving feed-backs from the model. These are inputs required for the 475 cooling calculations such as pressure, temperature, $CO_2$, $O(^3P)$ and other atmospheric constituent densities. It return the $CO_2$ 15 $\mu$m radiative cooling/heating at the altitude grid specified by the user. The routine works for day (SZA $\leq 110°$) and night (SZA $> 110°$) conditions.

Following the discussion in section 5 the routine may generally work with any upper and lower boundary, however it is nor recommended putting the upper boundary below $\sim 110$ km since it causes increasing calculations errors due to not accounting 480 for the upper atmospheric layers as well as placing the lower boundary below $\sim 20$ km because of increasing errors caused by not accounting for the line overlapping in a current version of the ODF approach.

The module requires *geometrical altitudes* to calculate radiative transfer and an *equidistant altitude grid*, which guaranties exact solution of the radiative transfer equation. The user may define any grid step including very fine one, which allows resolving micro-scale temperature disturbances. This is an advantage of our routine since its calculation time only linearly 485 depends on the number of grid points $N_D$. Compared to this the calculation time of matrix algorithms $\sim N_D^3$, see expressions (4 -10). Nevertheless, for those, who wants to account for additional cooling effect of the micro-scale sub-grid disturbances, we recommend to use its parameterization described in (Kutepov et al., 2007; Kutepov et al., 2013b), which may be easily implemented in the routine.

The routine includes all inputs required for its proper performance, among them all collisional rate coefficient parameter- 490 izations as described by Shved et al. (1998) as well as the HITRAN-2016 spectroscopic data for all bands available for the $CO_2$ level set in Table 1. The latter are presented as temperature dependent $A(T)$ and $B(T)$ Einstein coefficients for each band branch calculated in accordance with (Kutepov et al., 1998; Gusev and Kutepov, 2003). We compared $A(T)$ and $B(T)$ with those calculated for 2 earlier HITRAN versions and found the differences less than 0.1-0.2% for bands included in our $CO_2$ night time model. For some hot 15, 4.3-$\mu$m, and combinational bands included in the daytime model, these differences are of 495 the order of 0.5%, since data for these bands slightly vary from one HITRAN version to another.

The routine includes also the detailed table of basic ODF for a band branch in a broad ranges of temperature and pressure variations, which then re-scaled for calculating the radiative transfer in any individual band branch.

The supplied set of levels and spectral band information ensures the cooling/heating calculation errors for day and night to be below 1 K/day for any temperature disturbances. For smooth temperature profiles, the calculation errors are around 0.1-0.3 K/day (for 400 ppmv of $CO_2$).

Finally, the routine allows the user to switch on and off the vibrational levels and/or bands used in the model. This removing or adding vibrational levels will also automatically add (or removes) the bands related to these levels. If the user task can tolerate larger errors, the calculation speed can be increased at the cost of lowering the accuracy.

*Code availability.* The current version of the routine code is available at https://doi.org/10.5281/zenodo.8005028

*Author contributions.* Within more than 20 year of collaboration of both authors on the development of the ALI-ARMS code, which optimized version NLTE15$\mu$mCool-E is presented here, AK contributed most in the development of the ALI and ODF methodologies, he also wrote the manuscript draft supported by AF. AF contributed most to the development of the ALI-ARMS physical model, designed and implemented in the code the ODF based routines for radiative transfer treatment, performed all calculations presented in this paper as well as the detailed analysis of the routine computational performance.

*Competing interests.* The contact author has declared that none of the authors has any competing interests.

*Acknowledgements.* AK and AF would like to express deep gratitude to friends and colleagues who have made an invaluable contribution to the development of the ALI-ARMS code and promoted its applications and who unfortunately have already left this world: David Hummer (†2015), who provided an advanced stellar atmosphere non-LTE code and invaluable support with its adaptation for treating molecular bands in the planetary atmospheres, Gustav Shved (†2020), Rada Manuilova (†2021) and Valentine Yankovsky (†2021), who provided crucial contributions into the non-LTE physical model development, Richard Goldberg (†2019), who stimulated the ALI-ARMS application to the analysis of the TIMED/SABER observations, Uwe Berger (†2019) who drew our attention to the need of accurate radiative cooling calculation in GCMs and motivated developing the routine presented in this study. AK and AF also wants to thank Rolf Kudritcki, who provided for them an opportunity to work in the 1990-2000s at Universitäts Sternwarte München and study advanced non-LTE techniques used in stellar atmosphere studies, Vladimir Ogibalov and Oleg Gusev, who provided significant contributions to the code software development, and Ivan Hubeny, who introduced to them the ODF-technique, which revolutionized the code performance. They are also thankful to Dan Marsh and Valery Yudin who provided inputs for testing the new routine presented here.

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
