# Peer review of "New Routine NLTE15 $\mu$ mCool-E v1.0 for Calculating the non-LTE CO2 15 $\mu$ m Cooling in GCMs of Earth's atmosphere"

_Geoscientific Model Development, 2023_

## Author Response (AR1)

***Replies to referee comments RC1 and RC2 on the scientific article "New Routine NLTE15μmCool-E v1.0 for Calculating the non-LTE $CO_2$ 15 μm Cooling in GCMs of Earth's atmosphere" by A. Kutepov and A. Feofilov***

***Alexander Kutepov and Artem Feofilov***

We would like to thank Reviewer 1 for his/her helpful comments and sincere desire to aid us to improve the manuscript.

We gratefully accept almost all technical comments and included corresponding changes in the manuscript (see below detailed replies).

We reply in detail here to general and specific comments of Reviewer 1. Unfortunately, we may not accept some changes of the text suggested by this referee because our concept of this paper is different from how he/she sees it.

OVERVIEW

The paper presents a new routine to calculate the Non-Local Thermodynamic Equilibrium (NLTE) cooling/heating for CO2 isotopes at 15um wavelength in the Mesosphere and Lower Thermosphere (MLT); the routine is called NLTE15umCool-E-v1.0. To sum up, the paper aims to show this routine as a parameterization alternative for General Circulation Models (GCMs) of Earth's atmosphere. The authors claim it is more accurate and faster than the previous parameterization.

*We believe that there is a misunderstanding here. We do not claim that our routine is both more accurate and faster than the previous parameterization. We state clearly that it is **slower** than Fomichev-1998 routine. But it is much more accurate and flexible.*

GENERAL COMMENTS

The new routine represents a significant innovation and will be an important option for climate models simulating ancient atmospheres or performing projections based on CO2 increase trending. Still, the new routine brings advantages, such as an expanded volume mixing ratio (VMR) range, achieving 4000 ppmv, or even higher accuracy for temperature computed between 80 - 100 km height (non-LTE region).

*Generally, it is an exact algorithm, which stems from our research code, so it may work with any CO2 VMR, for instance, for a Martian atmosphere, which consists 99% of CO2. It is a further development of our routine, which is since 2005 applied in the MPI MGCM (for Max Planck*

*Institute Marc General Circulation Model) of Hartogh et al, 2005, [please see more also our replies to L. Rezac's comments (Kutepov and Feofilov, 2023b)]. We write in the text about 4000 ppmv as about an upper limit only because we show routine errors only up to this $CO_2$ VMR.*

*Our routine is not limited by the altitude range 80-100 km. The non-LTE effects for the $CO_2$ emission are particularly strong above 100 km. Our routine provides an exact solution of the non-LTE problem in the entire range of altitudes prescribed by the user. The upper boundary can be put at any altitude above about 110 km assuming corresponding atmospheric model inputs.*

Despite the valuable scientific significance of the new methodology, the authors submitted the article without proofreading, which substantially affected my review. Overall, the manuscript introduction delivered an unusual approach at the beginning and along the other sections, where some crucial information was missed. In addition, the authors claim that the new combination of techniques (ALI + ODF) significantly reduced the time consumption during the simulations, as shown in Table 2 in the Ttot column. However, the manuscript doesn't provide the data and scripts used to compute each non-LTE technique comparison, making verifying the results impractical. Still, the authors compared the parameterizations KF23 and F98 with a reference model (WACCM6 CESM) but didn't explain why this model was chosen and what non-LTE parameterization is adopted within his source code. Another point is about the NLTE15µmCool-E v1.0 source code, where the users don't have a manual with instructions for users.

*The issues discussed by the referee in this paragraph were addressed in our preliminary reply to his/her comments (Kutepov and Feofilov, 2023a). As for the lack of proofreading in the original version, we apologize for this and we provide a new version, which has been checked for grammar and typographical errors.*

Due to the paper presentation demanding a substantial review and improvements, the authors should rewrite the paper to achieve the desired presentation quality and resubmit the manuscript as soon as possible. Anyway, I tried to perform a complete article review because I enjoyed the new routine purpose and intend to use it. Obviously, tt's not a proofread, but it will be useful to rewrite the paper. Below, I shared some specific comments. After that, I shared some minor comments. Then, I hope the authors resubmit the revised manuscript to the GMD journal.

*We are grateful to this reviewer for his/her good will, detailed reading the manuscript, and for correcting grammar inconsistencies and typos. We tried to improve the text following the suggestions if they do not contradict with our concept of this manuscript as a technical paper.*

SPECIFIC COMMENTS

*1) The introduction starts abruptly from the radiative heat rate (h), also called radiative flux divergence (Eq.1).*

The introduction is one of the most essential sections of a paper, where the authors bring an informative discussion within the article's scope. However, the current preprint introduction is

similar to a subchapter of a book, which can push the readers away due to the need for more context.

*We see the point, but we assume this time that our target reader is already prepared for this discussion, because it is very unlikely that the unprepared one will bother about the implementation of a complex code into a complex model. This manuscript is a technical paper, which suggests an advanced technical tool for the solution of a well-known problem of MLT cooling, which has been being discussed since the first publication of Paul Crutzen (Crutzen, 1970). In the revised manuscript we enhanced the introduction following the reviewer suggestion.*

The paper should present some essential information, but I had to get the omitted information by reading other articles. For example, in the review "Infrared Radiation in the Mesosphere and Lower Thermosphere: Energetic Effects and Remote Sensing (Feofilov and Kutepov, 2012)" I found more precise information that is supposed to be present in the submitted preprint. The same in Kutepov et al. (2007) (https://doi.org/10.1029/2007GL032392) and Kutepov et al. (2016) (https://doi.org/10.5194/amt-10-265-2017, 2017).

*Unfortunately, we cannot copy-paste our previous texts in this manuscript (nor paraphrase them – both are considered as self-plagiarism). It is normal when the reader, who develops the interest to the paper, follows the references to get more information. We thank the referee for this/her interest to our previous works which helped him to write his report in a most objective and comprehensive way, which we appreciate.*

It's worth mentioning that in section 2, the authors describe the historical progress of some important iteration techniques along the century XX (e.g., Lambda Iteration (LI), Curtis-Matrix (CM), Accelerated Lambda Iteration (ILA), etc.). Thus, this section approaches the main article's subjects using a well-written structure despite minor details. On the other hand, the introduction is one of the preprint's weak points; to get around it, the authors must begin the paper by discussing attractive and interesting topics, considering increasing the article's readability. For instance, the Mars EXpression Mission (MEX) results or citing articles discussing the high levels of CO2 in the primitive atmosphere (4000 ppmv; threshold of the NLTE). After that, the authors would introduce the specific content gradually to present the general article's goals in the penultimate paragraph.

Overall, I recommend moving the introduction (3 first paragraphs) to section 2 and writing more general paragraphs at the initial of the introduction (e.g., astrophysics, ancient atmosphere, SABER, PFS-MEX, etc.).

*Following the reviewer comments, we added more general paragraphs in the introduction and moved parts of previous introduction to a new section.*

*However, we are of the opinion that introducing the variety of the non-LTE problems in the introduction will not help to promote the application of our routine in GCMs. One needs to*

*develop the "feeling" that the correct non-LTE colling is important for the problem he/she solves. One usually starts with applying LTE cooling, then some simple non-LTE approximations, etc., and finally may come (or may not) to the need of a good non-LTE routine. The MPI MGCM developers appeal to us to give them a safe tool for the non-LTE cooling calculation only after they recognized they needed it to fit the observations. Please, see also the answer to the 1st specific comment.*

*Also, we believe that mentioning in the introduction the importance of the non-LTE for analysis of observations like those of SABER, PFS-MEX, etc, would be not appropriate, since we deal in this paper with the cooling or heating effect of internal radiation field rather than with modeling the radiation outgoing from the atmosphere. Our cooling routine is the optimized version of basic ALI-ARMS research code, which was successfully applied to the analysis of many observations. Therefore, we mention the most interesting application of this code in the section 3.5. stressing that in this code the radiative cooling/heating is just a by-product of the non-LTE problem solution. This facilitates its application for calculating the non-LTE coolin.*

**2) Presentation and content connections**

Please the authors must draw attention to the text's coherence and cohesion; once a well-written manuscript connects all the given information gradually. Another problem is the omitted nomenclature abbreviations, affecting the readability. For instance, the equation to compute the number of operations for the solution of radiating transfer (Nrad) is not declared as an abbreviation in the paper, only in Eq. 5. The same happens with the number of auxiliary operations (Naux). Ultimately, even though some readers can likely be familiar with this technical vocabulary, it's reasonable never to present abbreviations without declaring it properly before in the text, which occurs often in the manuscript (e.g., GCM, MLT, LIMA, ODF, VV, GRANADA, SABER, HITRAN, and others.).

*We made all required corrections of the text.*

**3) GMD journal technical instructions**

The document was prepared in disagreement with the technical instructions provided by the Copernicus/GMD journal. I recommend reading the author's guidelines and downloading the manuscript template file (Latex, Word, or R markdown). For example, the citation of Fomichev et al. (1993) is cited as Fomichev, Kutepov, Akmaev, and Shved (1993); it needs to be corrected. For articles with more than two authors, the citation needs to use the Latin acronym "et al". Otherwise, the readers could consider only the last name as a citation. For example, Fomichev et al. (1993) would be Shved (1993).

**Done.**

Another problem is that Tables 1 and 2 disagree with the technical instructions. Therefore, adjusting your tables before submitting the article again is necessary. Note that Table 2 can be combined to become only one.

*We updated both tables.*

The authors mention the unit of measurement for cooling rate in the document, sometimes adopting K/day and K/Day, but the correct is K/d. Please fix it.

*We checked several papers from various journals and some books, for instance the book by Lopez-Puertas and Taylor (2001). We found that various authors prefer using different cooling/heating rate unit presentations: K/Day, K/day, K day$^{-1}$ , etc. among them K/d it is perhaps the rearrest variant used. Therefore, we have chosen to use in our manuscript "K/day" as the most widely used and clearly red and to avoid misunderstanding, for example, reading K/d as K/decade, which is used in some papers dealing the climate change. We leave this issue open for the discussion with the journal's technical team.*

**4) Figure 1: CO2 vibrational levels diagram …**

Figure 1 is based on Feofilov and Kutepov (2012) but is slightly different. The first problem in Figure 1 is the abbreviations FH, SH, TH, and FB. I didn't find the abbreviation meaning in the document, but in Feofilov and Kutepov (2012) review, each one is declared: First Hot (FH), Second Hot (SH), Third Hot (TH), and Fundamental Band (FB). Another problem is the CO2 isotopes code 626, 627, 628, and 636. The authors should declare explicitly the isotopes just like in Feofilov and Kutepov (2012) in section 2.2.3 ("*... The isotopes are marked using the lower digit of the atomic weight: 16O 12C 16O corresponds to 626 ...*"). Additionally, the authors should declare and distinguish the main isotopes of the minor CO2 isotopes in the text section 2.

I realized that the HITRAN codes (10002 and 02201) are in different positions than Figure 7 in Feofilov and Kutepov (2012). Thus, my question is: does it affect something in the diagram?

Please include the missed unit on the left side of the figure: Energy (cm^-1).

The Figure 1 caption is identical to Figure 7 in Feofilov and Kutepov (2012). I recommend using other words to avoid plagiarism issues.

*We thank the reviewer for noting the mistake with levels positioning in this plot. We now reproduce correctly Figure 7 from Feofilov and Kutepov (2012) and have rewritten its cooption.*

*We also have given the explanation of isotope notification in the paragraph, which described the Table 1, where these notifications are used.*

*Also, we give now complete description of what are the fundamental, first and second bands in the text where these notifications appear the first time.*

**5) Figures legends**

Figures 4, 5, and 6 must contain a box legend declaring the latitude with your corresponding color, such as Figures 2 and 3.

*We explained in the caption of Figure 4 (which is also applied to Figures 5 and 6) that line colors correspond to the those in the legend of Figure 2.*

**6) Model evaluation**

The authors present the KF23 and F98 routines errors compared to a reference model (WACCM6 CESM). Only the graphical plots and interval range aren't enough to validate KF23. I recommend employing the *Root Mean Square Error* (RMSE) to show that KF23 is more accurate than F98.

*We do not see any point in showing RMS values for a set of profiles, like, for instance, Lopez-Puertas et al, 2023 do. Our goal was to show real errors of both routines, whereas presenting RMSs is a way to wash out the individual errors. For instance, RMS for 100 profiles, where each result has an error +/-10 K/Day will be 1 K/Day, but this does not tell the truth about the routine's accuracy.*

Well, I have some important questions:

1) Why did the authors use the WCACCM6 CESM as a reference Model?

*It seems a misunderstanding. We do not use WACCM as reference model, we only used its pressure, temperature, and composition outputs as inputs for comparisons to estimate the accuracy of our new routine and that of Fomichev et al, 1998 versus our detailed **reference non-LTE model.***

*We selected the WACCM outputs because this model is well established and widely used by the community. Generally, any pressure/temperature profile disturbed by strong waves will give the same results. These may be both p/T generated by modern GCMs, like we did, or those retrieved from ground base or space observations, or even artificial wavy p/T generated by the user. We put corresponding clarification in the revised text.*

2) What is the parameterization adopted within the reference model? Please declare it in the manuscript.

*Again, WACCM in this study is not a reference model, see reply to the p. 1) above. WACCM uses many physical parameterizations. For the $CO_2$ 15 μm cooling it applies the Fomichev et al, 1998 parameterization. We put this information in the revised manuscript.*

3) What are the advantages and/or disadvantages of the KF23 compared to the parameterization adopted in the reference model?

*The paper discusses in detail advantages and disadvantages of the KF23 compared to the Fomichev et al, 1998 algorithm.*

The manuscript does not mention anything about the WCACCM6 CESM. If possible, request additional information from Dan Marsh.

*We included the reference to the model description.*

**6) Comments about the NLTE15μmCool-E v1.0 source code**

The program can be compiled using a Makefile, but during the first attempt, the compilation fails due to a deprecated GCC flag "g77" (Line 53 of the Makefile). I solved it by updating it to "gfortran", which can recognize all previous GNU Fortran versions (77, 90, 95, etc.). After that, I tested the program, and at a glance, everything worked well. A minor issue is in the file main.f at line 124, where I needed to provide additional space for Pressure (P) and Temperature (T) strings; otherwise, the value number would remain print merged with P and T in the console.

*Thanks for reporting these issues. We fixed them in the updated version of the code.*

Additionally, I would like to change some parameters to run different simulations, but the code "read_parameters.c" does not provide the parameter CO2 VMR. Thus, having a namelist to set CO2 values easier would be interesting. For instance, I am setting from 400 to 4000 ppmv.

*We see the point, but the problem here is twofold – first, the code already receives the CO2 VMR vertical profile from the model and there's no need to replace it with a single value, knowing that CO2 is well-mixed up only to 70-75 km in Earth's atmosphere. Second, such a namelist will require either a recompilation or a reading routine. Both will decelerate the testing and/or running the code. We suggest the reviewer to change the input parameters including the VMRs of CO2 and O through the inputs of the interface.*

Another question is about instructions to install the routine and explain some technical aspects of the program, such as the role of the objects and libraries. Otherwise, it might be hard to couple the routine in climate models. I strongly recommend preparing a readme file.

*As for the libraries, we do not use something external, all the matrix operations and other functions are coded in the framework of our code. Regarding the objects, they are created during the compilation, but it's the main executable, which is called by the interface. For obvious reasons, we cannot foresee all possible cases of coupling of our routine to GCMs, but our experience with Martian GCM and WACCM tells us that this is doable with a little effort. We added a readme file explaining the installation procedure.*

MINOR COMMENTS

Line 3: Typo, replace "nigh" with "night". ***Done.***

Line 3: Cooling rate "K/Day -> "K/d" and so on along the text.

*Please see our reply to the comment in the section "3) GMD journal technical instructions"*

Line 13: "… with the opposite sign" replace by "… with the opposite sign:" ***Done.***

Line 17: "… LTE 15 um band cooling" replace by "… LTE 15 um band cooling:" ***Done.***

Line 21: "Declare what is the GCM abbreviation" ***Done.***

Line 22: where (Curtis and Goody, 1956; …) use (e.g., Curtis and Goody, 1956; …). You should do the same in other parenthesis examples along the text.

*We did it here in and in other places, where "e.g." is appropriate.*

Line 40-41: The citation format is wrong, change it for Fomichev et al. (1993). ***Done.***

Line 48: Please avoid using terms like "below" and "above" within the document. After the typesetting stage of the manuscript, the final version will modify the position of the paragraphs, equations, tables, and figures. ***Done.***

Line 54: Declare what is MLT. ***Done.***

Line 55: Declare what LIMA is. ***Done.***

Line 63: Such as the Fomichev et al. (1998) parameterization is called F98; it would be reasonable to call the Kutepov and Feofilov (2023) parameterization of KF23. Please consider adopting KF23 instead of KF2023. ***Done.***

Line 69: ARMS wasn't declared before in the introduction, therefore, it should be explained in the manuscript as a full nomenclature: Atmospheric Radiation and Molecular Spectra (ARMS). ***Done.***

Line 71: ODF means Opacity Distribution Function, but it wasn't declared before. ***Done.***

Line 74: Please consider putting the citation (Hubeny and Mihalas, 2015) at the end of the sentence, Line 76. ***Done.***

Line 82: Add a missed comma after "… non-LTE problem …" -> "… non-LTE problem, …"

In the same line, you should change "… that in case …" -> "… that in the case …". ***Done.***

Line 88: Put the citations in the final of the sentence. ***Done.***

Line 94: Gramma -> Change "However, the convergence of both algorithms depends, strongly on the way the local non-linearity is treated, see next section." By "However, the convergence of both algorithms depends strongly on how the local non-linearity is treated, see next section." ***Done.***

Line 99: What is GRANADA? Declare the nomenclature in the sentence. ***Done.***

Line 113: What is VV? Please declare VV as the intermolecular Vibrational-Vibrational. ***Done.***

Line 139: What do you are comparing in "This way of treating the radiative transfer is about 50-100 times than the 140 classic LBL approach". You mean "…more than the…", "…faster than then…". ?

*We edited the text to explain how we reach this acceleration factor.*

Line 147: Clarify how it can be standard and modified simultaneously.

*To make clear what we mean we changed this sentence. It is now: "The core of it is a standard CM algorithm, which the authors in this and their earlier publications prefer to called MCM (for Modified Curtis Matrix)."*

Line 150: … " generalized …" : remove the space at the beginning. ***Done.***

Line 152: Put the citation at the final of the sentence. ***Done.***

Line 160: ALI-ARMS should be declared before. ***Done.***

Line 163: What is PFS? Declare it in the text (Planetary Fourier Spectrometer) ***Done.***

Line 164: What is SABER? Declare it in the text (Sounding of The Atmosphere Using Broadband Emission Radiometer) ***Done.***

Line 175: What is VT? Declare it (Vibrational-Translational). ***Done.***

Line 176: in Lopez-Puertas and Taylor (2001), add DOI and ISBN in the references. ***Done.***

Line 179: Please, change "ro-vibrational" to "rotational-vibrational". ***Done.***

Line 207-209: Consider including 1) time for solving the radiative transfer (Trad) … 2) time for auxiliary (Taux) … 3) time for matrix inversions (Tinv). ***Done.***

Line 210: After (Kutepov et al., 1998) add ":" . Please do the same before other equations. ***Done.***

Line 211: Remove the comma at the final of the equation. Do the same in other equations. ***Done.***

Line 217-218: Please make sure to regard the architecture name of the processor. Usually, x86 is 32-bit, whereas x64 is 64-bit. Do the same in Line 279.

*We agree that the typical nomenclature of the processor's architecture is as suggested by the reviewer. But, these very processors are capable of operation both in 32-bit and in 64-bit*

*architecture, that is specified in the corresponding /proc/cpuinfo files: Architecture: x86_64, CPU op-mode(s): 32-bit, 64-bit. We compiled the code to use them in 64-bit mode.*
*To exclude possible ambiguities, we updated the text as follows: We performed this study at two different machines, with x86_64 Intel and Intel Xeon Gold processors operating at 2.2 and 2.5 GHz, respectively. We compiled the ALI-ARMS code to be used in 64-bit architecture with the help of a standard gcc compiler and we ran it on a single processor*

Line 219: declare the gcc compiler nomenclature: gnu compiler collection. ***Done.***

Line 226: I suggest changing "Whereas N is …. " to "Thus, whereas N is defined by the mathematical nature of the problem and the algorithm applied, the coefficient C may depend on many factors, such as the quality of programming, language used, operational system, interpreter, computer architecture and performance, etc." ***Done.***

Line 237: It is worth remembering NL meaning (number CO2 vibrational levels) once the readers see NL declared only in line 23 on the second page. ***Done.***

Line 245: Change to "…. is approximately N^3" and where is "Therefore," change to "Thus, we have the following equations:" ***Done.***

Line 253: Include DOI and ISBN of the Book Press et al. (2002). ***Done.***

Line 271: After compared to the nighttime… include a comma. ***Done.***

Line 286: What is the reference model? Please, declare your name as well as justify the reasons to use it. In addition, declare the abbreviation Volume Mixing Ratio (VMR) here; otherwise, this abbreviation will appear suddenly at line 307. ***Done.***

 Line 289: Change the word above to before. ***Done.***

Line 290: What do you mean about "various waves"? If you are talking about gravity waves, declare it explicitly in the paragraph.  !!!!

*We added the corresponding explanation*

Page 13: In Figure 3, there is a typo in the caption: Solis, change to Solid.  ***Done.***

Line 319-321: Update the sentence to "For instance, the test for the nighttime for a roughly twice smaller set of bands, which does not include weak first and second hot bands of 626 and 636 isotopes, shows that the maximal cooling rate error for 400 ppmv may increase up to 3 K/d; however, computing time becomes only 10% shorter (see also Table 2)." ***Done.***

Line 333: include a comma after "… absorption and assimilation" ***Done.***

Line 335: You don't declare before what is VT and VV. ***Done.***

Line 370: The authors say, "… many previous studies (e.g., Lopez-Puertas and Taylor (2001)). I was expecting to cite at least three papers. Add more citations or modify "… many previous studies". **Done.**

Line 377-378: Consider changing the sentence to "The accuracy tests of the KF23 routine were performed for a 1 km step grid with the upper boundary of the atmosphere at 130 km." **Done.**

Line 380: Add a comma after However  **Done.**

Line 395: Please consider updating the last conclusion paragraph to "The KF23 routine provides accurate cooling calculations in a vast range of k and O(3 395 P) variations. It also works well for very broad variations of CO2, both below and above the current density, up to 4000 ppmv. Consequently, this allows us to use this routine to model the Earth's ancient atmospheres and the climate changes caused by increasing CO2."

**Done.**

Line 424: Typo, replace "…. User to switch on an off" with "… User to switch on and off"?

**Done.**

OVERVIEW

The solid knowledge on the spatial-temporal distributions of $CO_2$ cooling/heating are desirable for modelling of dynamics and temperature in General Circulation Models. A number of parameterizations are used for this purpose. All of them are characterized by either lack of accuracy or high numerical and time costs. This article describes a model that is devoid of these disadvantages.

The research is scientifically valuable. The theoretical part is presented in the paper very convincingly. The methods and approaches are correct. On my opinion this work brings deep insights on modelling of MLT region and new model essential for precise and efficient calculations. On my opinion this work should be accepted in Geoscientific Model Development after minor revision.

*We are very grateful to this referee for recognizing the importance for MLT GCMs of a new routine we present, and for very interesting comments, which stimulated us to improve the manuscript text to make it more informative.*

**Comments.**

**1.** The ODF technique is explained only briefly, the authors refer to their previous works on the implementation of the ODF to the radiative transfer in molecular bands in the planetary atmospheres. But they say no word about the limitation of this technique - is it 100% equivalent to line-by-line in terms of accuracy? If not, what are the errors introduced by this technique? Do these errors depend on the pressure-temperature profile or on $CO_2$ concentration or both?

*We agree that the limitations of the ODF technique were not discussed in the paper. Indeed, the ODF is a very useful technical approach, which is not 100% equivalent to line-by-line in terms of accuracy. However, it introduces very small errors in the 15-micron cooling calculations. For current CO2 density (~400 ppm in the lower atmosphere), these errors do not exceed 0.3 K/Day in a broad range of temperature variations, see Figure 18 of Feofilov and Kutepov, 2012. They increase roughly linearly with the CO2 increase if the pressure is fixed. Additionally, with the pressure increase line overlapping needs to be accounted. We added corresponding paragraph in the revised text.*

*Since our current ODF approach is developed for the non-overlapping lines, we do not recommend using our routine below 20 km where overlapping becomes important.*

*We are preparing the paper about the ODF technique, where we describe in detail how it works applied to various problems. We do not show in this technical paper all accuracy tests, which will overload it. Instead, we show **cumulative** errors caused by ODF approach, reduced set of levels and bands, etc, since cumulative errors are most important for the end user.*

2. Additionally, the authors wrote that they apply ODF exceptionally for the band branches. It looks more reasonable to apply it directly to the entire band. May the authors explain why they do not do this?

*We do not apply ODF to the entire bands because our tests show that it reduces the accuracy of calculations, and there's a physical reason for this, which we describe in the paper in preparation. We show in it that the accuracy cannot be restored even if we double or triple the number of frequency points describing the ODF profile of entire band.*

3. How the final result is sensitive to the completeness of the spectral database used for an input? I understand that the code uses a pre-formatted HITRAN dataset, but does one need to reprocess this dataset for each new version of HITRAN?

*It is correct, we do not apply directly the HITRAN data, but, use, as we call it the "mini-HITRAN" data set. This set comes from the pre-calculated A and B Einstein coefficients for band branches. These coefficients were calculated as described by Kutepov et, al, 1998 and Gusev and Kutepov, 2003, using HITRAN-2016. We also compared them with those calculated for 2 earlier HITRAN versions and found the differences less than 0.1-0.2% for bands included in our CO2 night time model. For some hot 15, 4.3-micron, and combinational bands included in the daytime model, these differences are of the order of 0.5%, since data for these bands slightly vary from one HITRAN version to another. We do not believe that any further version of HITRAN or any other spectral database will significantly affect these values. Addressing this question, we added the information about the accuracy of spectral data we use.*

4. When authors write that the "accuracy is not sacrificed", when they change their conversion criterion from 1e-4 to 1e-2, what exactly do they mean? Could you, please, be more specific and provide actual numbers and indicate the test conditions?

*First of all, we define the convergence at each iteration as a ratio of the level's population change to the level's population, checked over all altitudes and all vibrational levels. For a given iteration, the worst converged level/altitude defines the convergence value. The non-LTE iterations stop and we consider the problem to be converged when the convergence value is below the convergence threshold. Answering the question we'd like to explain that we performed a series of test calculations, varying the non-LTE problem convergence threshold, and did not see a significant change in cooling/heating cumulative errors for any of the atmospheric profiles we used if the convergence criterion changes from 1e-4 to 1e-2.*

5. The model is supposed to work in a plane-parallel approach. What are the errors associated with abandoning spherical geometry? Are they different for the nighttime and daytime?

*Lopez-Puertas and Taylor, 2003 stated that plane-parallel geometry is a very good approximation for the middle and upper atmosphere by the solution of the non-LTE problem. In our case perhaps only one concern related to the sphericity of the atmosphere is the absorption of solar radiation and related heating for large solar zenith angles. However, the routine accounts explicitly the atmospheric sphericity by calculating the solar radiation impact.*

References

Crutzen, P. J.: Discussion of paper "Absorption and emission by carbon dioxide in the atmosphere" by J. T. Houghton, Quart. J. Roy. Met. Soc., 96, 767–770, 1970.

Feofilov, A. G. and Kutepov, A. A.: Infrared Radiation in the Mesosphere and Lower Thermosphere: Energetic Effects and Remote Sensing, Surveys in Geophysics, 33, 1231–1280, https://doi.org/10.1007/s10712-012-9204-0, 2012.

Gusev, O. A. and Kutepov, A. A.: Non-LTE Gas in Planetary Atmospheres, in: Stellar Atmosphere Modeling, edited by Hubeny, I., Mihalas, D., and Werner, K., vol. 288 of Astronomical Society of the Pacific Conference Series, p. 318, 2003.

Hartogh, P., A. S. Medvedev, T. Kuroda, R. Saito, G. Villanueva, A. G. Feofilov, A. A. Kutepov, and U. Berger, Description and climatology of a new general circulation model of the Martian atmosphere, J. Geophys. Res., 110, E11008, doi:10.1029/2005JE002498, 2005.

Kutepov, A.A., and Feofilov A.F., "Reply on Comment on gmd-2023-115 of referee 1", https://doi.org/10.5194/gmd-2023-115-AC1, 2023a.

Kutepov, A.A., and Feofilov A.F., "Reply on Comment on gmd-2023-115 of Ladislav Rezac", https://gmd.copernicus.org/#AC2, 2023b.

Kutepov, A. A., Gusev, O. A., and Ogibalov, V. P.: Solution of the non-LTE problem for molecular gas in planetary atmospheres: superiority of accelerated lambda iteration., Journal of

Quantitative Spectroscopy and Radiative Transfer, 60, 199–220, https://doi.org/10.1016/S0022-4073(97)00167-2, 1998.

Lopez-Puertas, M. and Taylor, F. W.: Non–LTE radiative transfer in the atmosphere, Singapore: World Scientific, ISBN 9810245661, DOI: 10.1142/9789812811493, 2001.